# Single-Cell and Bulk RNA Sequencing Reveal Malignant Epithelial Cell Heterogeneity and Prognosis Signatures in Gastric Carcinoma

**DOI:** 10.3390/cells11162550

**Published:** 2022-08-17

**Authors:** Zhihong Huang, Chao Wu, Xinkui Liu, Shan Lu, Leiming You, Fengying Guo, Antony Stalin, Jingyuan Zhang, Fanqin Zhang, Zhishan Wu, Yingying Tan, Xiaotian Fan, Jiaqi Huang, Yiyan Zhai, Rui Shi, Meilin Chen, Chunfang Wu, Huiying Li, Jiarui Wu

**Affiliations:** 1Department of Clinical Pharmacology of Traditional Chinese Medicine, School of Chinese Materia Medica, Beijing University of Chinese Medicine, Beijing 100029, China; 2School of Life Sciences, Beijing University of Chinese Medicine, Beijing 100029, China; 3School of Management, Beijing University of Chinese Medicine, Beijing 100029, China; 4Institute of Fundamental and Frontier Sciences, University of Electronic Science and Technology of China, Chengdu 610054, China; 5Department of Operations, Beijing Zest Bridge Medical Technology Inc., Beijing 100176, China; 6School of Biology, Beijing Forestry University, Beijing 100091, China

**Keywords:** single-cell RNA sequencing, bulk RNA sequencing, gastric carcinoma, malignant epithelial cells, prognostic signatures

## Abstract

Gastric carcinoma (GC) heterogeneity represents a major barrier to accurate diagnosis and treatment. Here, we established a comprehensive single-cell transcriptional atlas to identify the cellular heterogeneity in malignant epithelial cells of GC using single-cell RNA sequencing (scRNA-seq). A total of 49,994 cells from nine patients with paired primary tumor and normal tissues were analyzed by multiple strategies. This study focused on the malignant epithelial cells, which were divided into three subtypes, including pit mucous cells, chief cells, and gastric and intestinal cells. The trajectory analysis results suggest that the differentiation of the three subtypes could be from the pit mucous cells to the chief cells and then to the gastric and intestinal cells. Lauren’s histopathology of GC might originate from various subtypes of malignant epithelial cells. The functional enrichment analysis results show that the three subtypes focused on different biological processes (BP) and pathways related to tumor development. In addition, we generated and validated the prognostic signatures for predicting the OS in GC patients by combining the scRNA-seq and bulk RNA sequencing (bulk RNA-seq) datasets. Overall, our study provides a resource for understanding the heterogeneity of GC that will contribute to accurate diagnosis and prognosis.

## 1. Introduction

Gastric carcinoma (GC) has a high incidence and mortality globally, and the most common histological type is stomach adenocarcinoma (STAD) [1,2]. GC is characterized by particular subtypes and clinical manifestations, which are the great challenges for clinical diagnosis and treatment [3,4]. Although advances have been made in identifying different molecular subtypes of GC through The Cancer Genome Atlas (TCGA) and Asian Cancer Research Group (ACRG), the prognosis improvement of patients is not significant [5,6]. GC is mainly the pathological change of gastric mucosa glandular epithelial cells, and the exact pathogeny is unclear. This study shows that *H. pylori* infection, a high salt diet, and a low intake of fruits and vegetables are risk factors for GC [7].

High-throughput sequencing technologies, such as bulk RNA sequencing (bulk RNA-seq) and single-cell RNA sequencing (scRNA-seq), have greatly accelerated the pace of research into the molecular characteristics of human tumors. So far, a large number of omicsstudies have described the GC’s genetic and epigenetic heterogeneity [8,9,10]. Although bulk RNA-seq can provide data for mass gene expression profiling, it cannot distinguish the relationship between different cell lineage and cellular interaction [11]. The advent of scRNA-seq compensates for the shortcomings of bulk RNA-seq. It provides a method that can characterize the transcriptional status of thousands of cells simultaneously [12,13]. The integration of scRNA-seq and bulk RNA-seq is of great scientific importance for the study of tumor development and heterogeneity.

Here, we constructed a transcriptomic landscape of GC malignant epithelial cells by scRNA-seq, encompassing various subtypes of GC pathology subtypes. We also combined the results with bulk RNA-seq to find the prognostic signatures that can predict the overall survival (OS) of GC patients. The landscape of GC malignant epithelial cells was constructed to interpret the heterogeneity of GC and shed light on the clinical diagnosis and prognosis of GC in our study (Figure 1).

## 2. Methods

### 2.1. Data Source and Preprocessing

The GC scRNA-seq dataset GSE183904 was downloaded as required from the Gene Expression Omnibus (GEO) database (https://www.ncbi.nlm.nih.gov/gds, accessed on 24 March 2022). The nine patients with paired primary tumor and normal tissues were selected for analysis in the present study. Using the R package Seurat, all samples were analyzed for genes/features shared by more than three cells and for cells containing 500–6000 features. Cells with mitochondrial RNA content higher than 20% were excluded [14].

Public clinical data and gene expression information were obtained from the TCGA database (https://portal.gdc.cancer.gov/, accessed on 24 March 2022), the Genotype-Tissue Expression (GTEx) database (https://www.gtexportal.org/, accessed on 24 March 2022) and the GEO database [5]. The bulk RNA-seq dataset included TCGA-STAD, GTEx-stomach, GSE15459, GSE29272, GSE57303, GSE62254, and GSE66229. Differentially expressed genes (DEGs) were calculated using the R package limma between tumor and normal samples. Only genes with an expression fold change (FC) >2.0 or <−2.0 and a false discovery rate (FDR) < 0.05 were taken into subsequent analysis.

### 2.2. scRNA-Seq Normalization, Clustering and DEGs Analyzing

SCtransform normalization was carried out separately for each sample, and FindIntegrationAnchors was run to find anchor genes and integrate the data [15]. The integrated data were scaled, principal component analysis (PCA) was performed, and the data were visualized using the t-Distributed stochastic neighbor embedding (tSNE) method. A shared nearest neighbor (SNN) modularity optimization-based clustering algorithm with a resolution of 1.0 was set to identify cell clusters. The FindAllMarkers module was used to identify DEGs, and genes expressed in more than 25% of cells in each cluster were selected for cluster demarcation. The cell type markers were employed to construct cell atlas.

### 2.3. Recognition of Malignant and Non-Malignant Epithelial Cells

To identify chromosomal copy number variations, InferCNV (https://github.com/broadinstitute/inferCNV, accessed on 28 March 2022) was used to explore scRNA-seq datasets from primary tumors compared with normal scRNA-seq datasets. Each epithelial cell in the present study was clustered, and a CNV score was calculated using the k-means clustering algorithm. The classes with a CNV score higher than the average was defined as a group of malignant epithelial cells, whereas the others were considered as a group of non-malignant epithelial cells. The malignant epithelial cells were also clustered and annotated as different subtypes based on cell type markers. DEGs between malignant and non-malignant epithelial cells and among different subtypes of malignant epithelial cells were identified using the FindAllMarkers module.

### 2.4. Functional Enrichment Analysis

Gene ontology (GO) enrichment analysis, Kyoto Encyclopedia of Genes and Genomes (KEGG) pathway enrichment analysis, and gene set enrichment analysis (GSEA) were performed to uncover the potential biological function of the different cells through the Database for Annotation, Visualization, and Integrated Discovery (DAVID, https://david.ncifcrf.gov/, accessed on 28 March 2022) database and the Molecular Signatures Database (MSigDB, https://www.gsea-msigdb.org/, accessed on 28 March 2022). The enrichment results were visualized using the R package ggplot2.

### 2.5. Trajectory Analysis

Pseudotime and cell trajectory analyses were used to investigate the differentiation trajectories and related genes in different cell clusters and to explain the molecular mechanism in the progression of GC. The R package Monocle was used to cluster cells and draw learning maps to compare trajectories among different subtypes of malignant epithelial cells [16,17].

### 2.6. Cell–Cell Communication Analysis

To explore underlying interactions between various subtypes in malignant epithelial cells, an analysis of cell–cell communication was conducted using the R package Cellchat, which simulates intercellular communication by binding ligands, receptors and their cofactors [18]. The enriched receptor-ligand interactions between the two cell types were deduced based on the receptor expression of one cell type and the corresponding ligand expression of another cell type.

### 2.7. Construction of Gene Regulatory Network

SCENIC is a tool for simultaneously reconstructing gene regulatory networks and identifying stable cell states from scRNA-seq data [19]. The gene regulatory network was derived based on co-expression and DNA motif analysis, and then the network activity in each cell was analyzed to identify the cell state. In order to determine the search space around the transcription start site, two gene-motif rankings (10 kb around the transcription start site or 500 bp upstream and 100 bp downstream of the transcription start site) were used as reference. Gene regulation of various malignant epithelial cells was constructed using the R package GENIE3, RcisTarget and AUCell. Then, the gene regulatory network was visualized using Cytoscape software.

### 2.8. Generation and Validation of the Prognostic Signatures

The bulk RNA-seq datasets GSE15459, GSE57303 and GSE66229 from the same platform were integrated for subsequent analysis. The DEGs from the scRNA-seq and the bulk RNA-seq datasets were screened related to the OS of TCGA-STAD patients using a univariate Cox analysis. Moreover, LASSO analysis was employed to choose the reliable predictors for multivariate Cox regression. Then, the prognostic signatures were used to create a polygenic risk score and divide the TCGA-STAD samples into low- or high-risk groups. The time-dependent receiver operating characteristic (ROC) curve was used to assess the predictive ability of the prognostic signatures. The GSE62254 dataset was used to validate the prognostic value of the prognostic signatures. All results were analyzed and visualized using the R package survival, survminer, rms and timeROC.

### 2.9. Statistical Analysis

R version 4.0.3 was used for statistical analysis (https://www.r-project.org/, accessed on 28 March 2022). The Wilcoxon-rank sum test was used to evaluate associations with continuous variables. Student’s *t*-test was used to analyze significant differences among distinct groups. Kaplan–Meier curves with log-rank statistics were used to compare OS. *p* < 0.05 was considered to indicate statistical significance.

## 3. Results

### 3.1. A Single-Cell Transcriptome Atlas of GC

Cell populations and corresponding molecular characteristics in GC were investigated in the GSE183904 dataset, which includes nine patients with paired primary tumor and normal tissues (Appendix A). After the removal of low-quality cells based on the screening criteria in Methods 2.1, 49,994 cells were persisted for biological analysis (Figure 2A). The cells were divided into 33 clusters after normalization, integration and PCA. These clusters were noted as eight known cell types according to marker genes recorded in the literature. The details were as follows: (1) the epithelial cells highly expressing *CDH1*; (2) the endothelial cells highly expressing *PLVAP*; (3) the fibroblast cells highly expressing *FN1*; (4) the T cells highly expressing *CD8A*; (5) the B cells highly expressing *TNFRSF17*; (6) the macrophage cells highly expressing *CD163*; (7) the NK cells highly expressing *KLRD1*; (8) the mast cells highly expressing *KIT*. The proportion of each cell type varied in different samples (Figure 2B–E, Appendix A).

### 3.2. Classification of Malignant and Non-Malignant Epithelial Cells

First, the cell population annotated as epithelial cells in the dataset was accurately determined and clustered (Figure 3A). Next, InferCNV was employed to distinguish malignant and non-malignant epithelial cells according to the CNV score. The kemans classes with a CNV score higher than the average (0.00124) were defined as the malignant epithelial cells group, and the others were defined as the non-malignant epithelial cells group (Figure 3B–E, Figure 4A). The top five DEGs of malignant and non-malignant epithelial cells were as follows: (1) malignant epithelial cells highly expressing *CAPN8*, *CLDN4*, *CYP3A5*, *PHGR1*, and *PLEC*; (2) non-malignant epithelial cells highly expressing *IGFBP2*, *LIPF*, *PGA3*, *PGA4*, and *PGA5* (Figure 4B,C). Compared with non-malignant epithelial cells, malignant epithelial cells were enriched for signaling pathways such as TNF-α/NF-κB, KRAS and IL6/STAT3/JAK. Moreover, there were also cells enriched in the epithelial mesenchymal transition genes associated with cancer development and progression (Figure 5A, Appendix A).

### 3.3. Transcriptional Heterogeneity of Malignant Epithelial Cells

The malignant epithelial cells were accurately determined and clustered. Subsequently, the subtypes have been annotated based on marker genes in the literature. There were three subtypes as follows: (1) the pit mucous cells with high expression of *GKN1*, *GKN2*, *MUC5AC*, and *TFF1*; (2) the chief cells with high expression of *PGC*; (3) the gastric and intestinal cells with high expression of *TFF1*, *TFF3* and *REG4* (Figure 5B–D, Appendix A). The proportion of each subtype varied considerably among the different Lauren’s classifications, and the top 20 DEGs of the three malignant epithelial cell subtypes were shown (Figure 6A,B). The GSEA results show that the pit mucous cells were mainly enriched in the pathways and gene sets correlated with cell proliferation, metastasis and invasion, cell differentiation and inflammatory response. The chief cells were mainly enriched in the pathways and gene sets related to cell proliferation, cell migration and invasion, cell cycle, cell differentiation, inflammatory response and cell survival. The gastric and intestinal cells were enriched mainly in pathways and genes that were relevant to apoptosis, cell growth, cell survival and cell metabolism (Figure 6C, Appendix A).

### 3.4. Trajectory of Malignant Epithelial Cells

Trajectory analysis was performed to identify the differentiation heterogeneity of malignant epithelial cells within and between tumors. It was found that the underlying cell differentiation trajectories of malignant epithelial cells comprised nine states, and the contents of three subtypes differed in various states. The pit mucous cells differentiated into the chief cells and the gastric and intestinal cells (Figure 7A). In the whole potential differentiation trajectories of malignant epithelial cells, the activities of genes related to the biological processes (BP) and pathways of cell proliferation, cell migration and invasion, cell cycle and apoptosis were reduced. In contrast, the activities of genes associated with translation, enzyme activity and regulation of other cell biological processes and pathways were increased (Figure 7B, Appendix A).

### 3.5. Intercellular Communication in Malignant Epithelial Cells

Cellchat was used to identify ligand–receptor pairs and molecular interactions among the three subtypes of malignant epithelial cells. The results show that multiple ligand–receptor-mediated cell interactions existed mainly in the MK signaling pathway (MDK-SDC4, MDK-SDC1, MDK-NCL, MDK-LRP1, MDK-ITGA6+ITGB1 and MDK-ITGA4+ITGB1) and in the MIF signaling pathway (MIF-CD74+CXCR4, MIF-CD74+CXCR2 and MIF-CD74+CD44) (Figure 8A–C, Appendix A).

### 3.6. Construction of Gene Regulatory Network in Malignant Epithelial Cells

The enrichment results of transcription factor (TF) in three subtypes of malignant epithelial cells were displayed, and the top 5 TFs were selected for further study (Figure 9A–C, Appendix A). The top 5 TF-gene regulatory networks were visualized using Cytoscape software (Figure 10A). The enrichment analysis results show that the common genes regulated by the top 5 TFs in three subtypes of malignant epithelial cells were mainly related to cell proliferation, signal transduction, and metabolism. The special genes regulated by the top 5 TFs in pit mucous cells were enriched in pathways and biological processes associated with cell proliferation, signal transduction and metabolism. The special genes regulated by the top 5 TFs in the chief cells were involved in the cell cycle and cell survival. The special genes regulated by the top 5 TFs in the gastric and intestinal cells involved the cell migration, invasion, and translation process (Figure 10B, Appendix A).

### 3.7. Bulk RNA-Seq Data Analysis

Differential gene expression analysis of all bulk RNA-seq datasets included in the present study was used to identify the DEGs between GC tumors and normal tissues. Then, the intersection of the results was taken to extract the common up-regulated and down-regulated genes for functional enrichment analysis (Figure 11A,B). The down-regulated genes were mainly enriched in the biological processes and pathways related to metabolism and normal gastric function. In addition, the up-regulated genes were involved in the behavior of malignant tumor cells, such as cell migration and invasion (Figure 11C, Appendix A).

### 3.8. Generation and Validation of the Prognostic Signatures in Malignant Epithelial Cells

The up-regulated genes from bulk RNA-seq and DEGs in malignant epithelial cells from scRNA-seq were integrated to construct a polygenic risk score based on prognostic signatures for predicting prognosis in GC. Ten genes (*AKR1B1*, *CFDP1*, *IMPACT*, *PRR15L*, *PTTG1IP*, *SLC17A9*, *STX10*, *TRIM25*, *UPP1* and *VCAN*) were selected to calculate the polygenic risk score through univariable, multivariable Cox and LASSO regression. GC patients were divided into low-risk or high-risk groups, and the prognosis of patients in the high-risk group was poor in the training cohort (TCGA-STAD) and test cohort (GSE62254). The ROC curve showed that the ten prognostic signatures performed well in predicting the OS of GC patients, with an area under the ROC curve > 0.7 in the training cohort and around 0.6 in the test cohort (Figure 12A,B). Prognostic signatures in three subtypes of malignant epithelial cells were also generated and validated with prognostic values for GC patients (Appendix A).

## 4. Discussion

GC is a heterogeneous disease influenced by many factors, presenting a number of difficulties for clinical diagnosis and individualized treatment. Burgeoning scRNA-seq has been widely applied to explore tumor heterogeneity, including the analysis of tumor development, drug resistance programs, cell–cell communication and immune infiltration patterns [20,21,22]. In the present study, we used the technology to construct a comprehensive landscape of malignant epithelial cells from GC at single-cell resolution.

Focusing on malignant epithelial cells, we identified three subtypes (the pit mucous cells, the chief cells, and the gastric and intestinal cells) with different transcriptome characteristics. The pit mucous cells and the chief cells were mainly derived from diffuse and intestinal GC patients, respectively. The gastric and intestinal cells had the largest proportion in GC patients with mixed intestinal and diffuse conditions. This suggests that the different Lauren’s histopathology of GC might originate from various subtypes of malignant epithelial cells. We can use the marker genes of different subtypes of malignant epithelial cells to help determine the Lauren’s classification. The results of the functional enrichment analysis show that they had different activities in the signaling pathways and biological processes associated with tumor development. The pit mucous cells and the chief cells were mainly enriched in cell proliferation, differentiation, migration and invasion, while the gastric and intestinal cells focused on cell survival and metabolism. The trajectory analysis results suggest that the differentiation of malignant epithelial cells could occur from pit mucous cells to chief cells and then to gastric and intestinal cells. These findings suggest that the tumor cells of diffuse-type GC were poorly differentiated and the cases with intestinal histology had a different degree of differentiation [23]. Zhang et al. found that the GC with a low differentiation degree was more aggressive, which was consistent with our findings [24].

Moreover, the cell–cell interactions indicated that the different subtypes of malignant epithelial cells communicate with each other to a high degree. The cytokines mainly mediated the intercellular communication among the three subtypes of malignant epithelial cells in the MK and MIF signaling pathway, which are involved in the inflammatory response, tumor growth, and cell migration, which have been poorly studied in GC [25,26,27]. The gene regulatory network in malignant epithelial cells showed that TF and its activities varied in different subtypes of malignant epithelial cells. Thus, the three subtypes of malignant epithelial cells were enriched in the different biological processes and pathways associated with malignant biological behavior. The above results suggest that intratumoral and intertumoral heterogeneity is a fundamental property of GC malignant epithelial cells.

In this study, we integrated the scRNA-seq and bulk RNA-seq datasets to find the prognostic signatures for predicting OS in GC patients. The prognostic signatures appeared to be essential for GC because they performed well in the train and test cohorts and could be used for GC clinical screening. In addition, we also generated and validated the prognostic signatures for different subtypes of malignant epithelial cells. They had prognostic value for the patients with different malignant epithelial cells in combination with Lauren’s histopathology in clinical use. We could predict the progression of patients by detecting the expression of these prognostic signatures.

The results of this study may be used in clinical diagnosis and treatment in the future. The cell atlas and marker genes of GC malignant epithelial cells can be applied to identify Lauren’s histopathology of GC patients by combining diagnostic methods such as endoscopy. Developing personalized therapy for malignant epithelial cell heterogeneity to improve clinical efficacy and the prognostic signatures can be used to test clinical application value and design the targeted drugs. At the same time, this study also had some limitations. First, the number of patients included in the study was small, and it was impossible to determine whether there was the same result in large-scale samples. Second, all results of this study were not validated in experiments and clinical practice to ensure they can be extended.

## 5. Conclusions

In summary, using scRNA-seq, we constructed a single-cell transcriptome atlas of GC malignant epithelial cells, encompassing several GC Lauren’s histopathological subtypes. Various methods analyzed the subtypes of malignant epithelial cells, and the heterogeneity in terms of functional enrichment, cell differentiation trajectory, and intercellular communication was preliminarily clarified. Furthermore, we combined the scRNA-seq with the bulk RNA-seq datasets to find the prognostic signatures for predicting the OS of GC patients. All the results of the present study need to be further validated in experimental and clinical practice.

## Figures and Tables

**Figure 1 cells-11-02550-f001:**
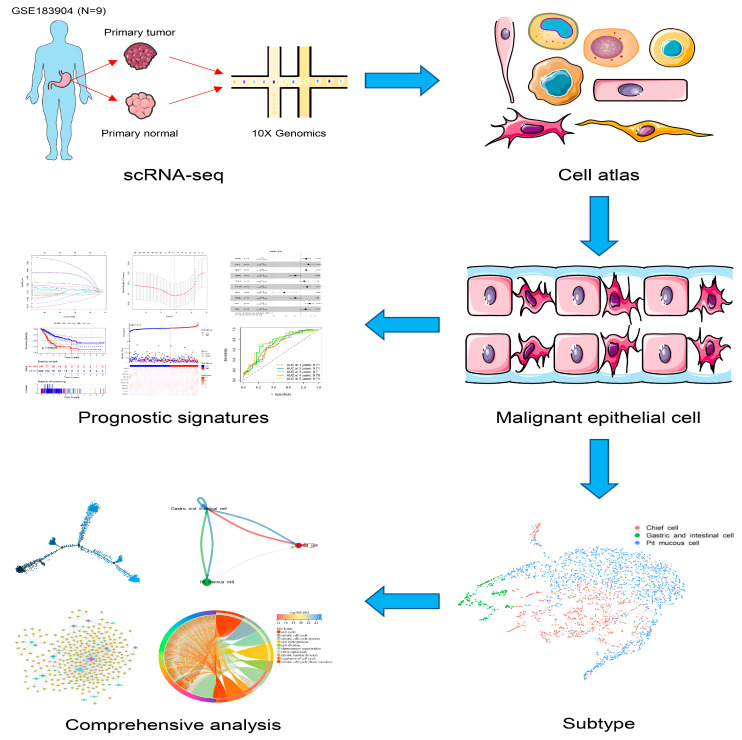
The workflow of the present study.

**Figure 2 cells-11-02550-f002:**
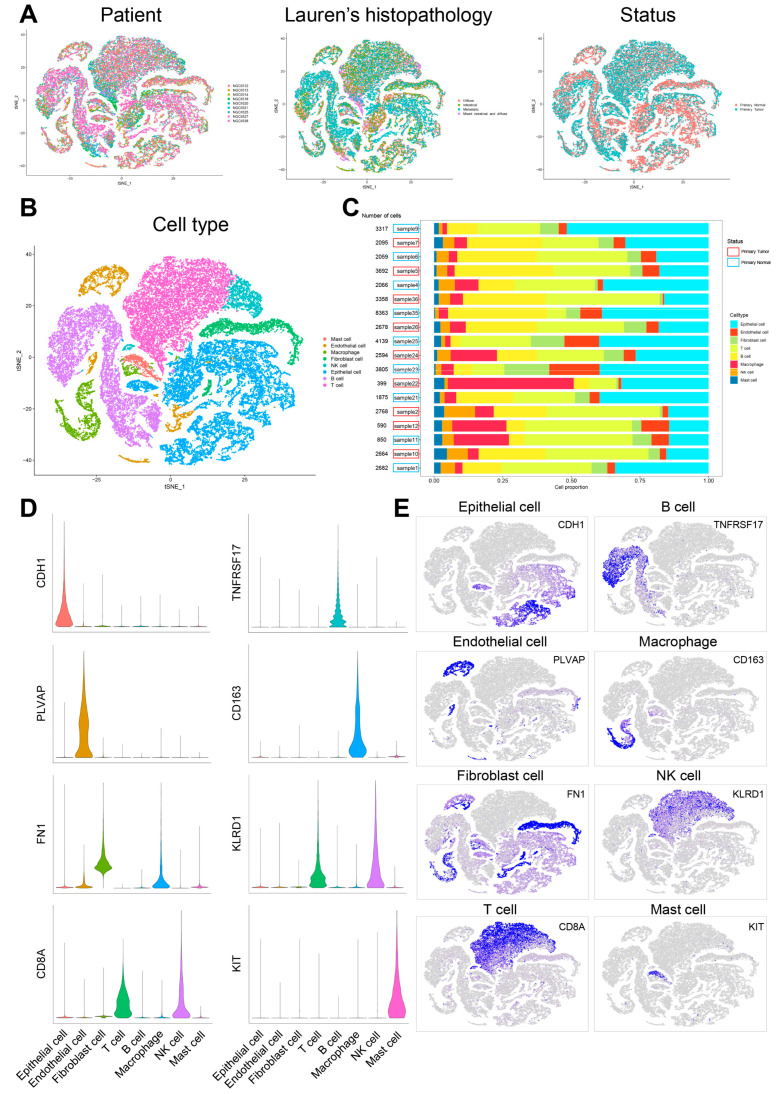
Cellular atlas of GC primary tumor and normal tissues. (**A**) tSNE plot for the 49,994 high-quality cells showing sample origin, Lauren’s histopathology and status. (**B**) tSNE plot showing cell types for the 49,994 cells. (**C**) The proportion of each cell type in 18 samples. (**D**) Violin plots showing the expression distribution of marker genes in eight cell types. (**E**) tSNE plots showing the expression levels of marker genes for eight cell types.

**Figure 3 cells-11-02550-f003:**
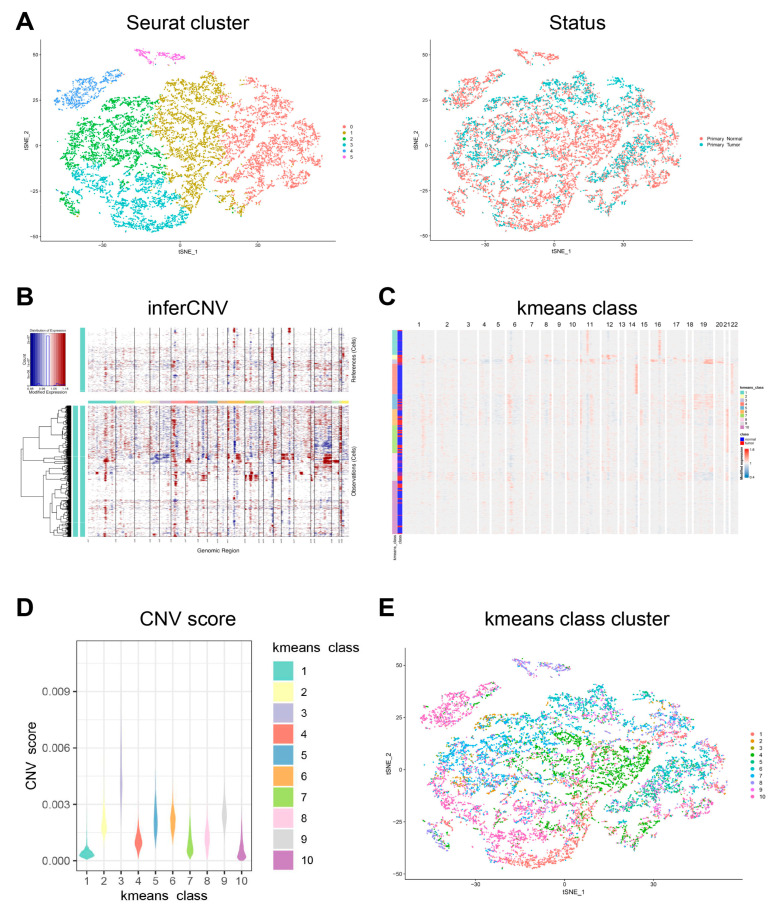
Copy-number variation analysis of epithelial cells. (**A**) tSNE plots showing the Seurat cluster and status of epithelial cells. (**B**) Hierarchical heatmap showing large-scale CNVs in the epithelial cells of a primary tumor and in normal tissues. (**C**) Hierarchical heatmap showing the results of k-means class clustering. (**D**) Violin plots showing CNV scores in the different k-means classes. (**E**) tSNE plot showing the epithelial cells in the different k-means class clusters.

**Figure 4 cells-11-02550-f004:**
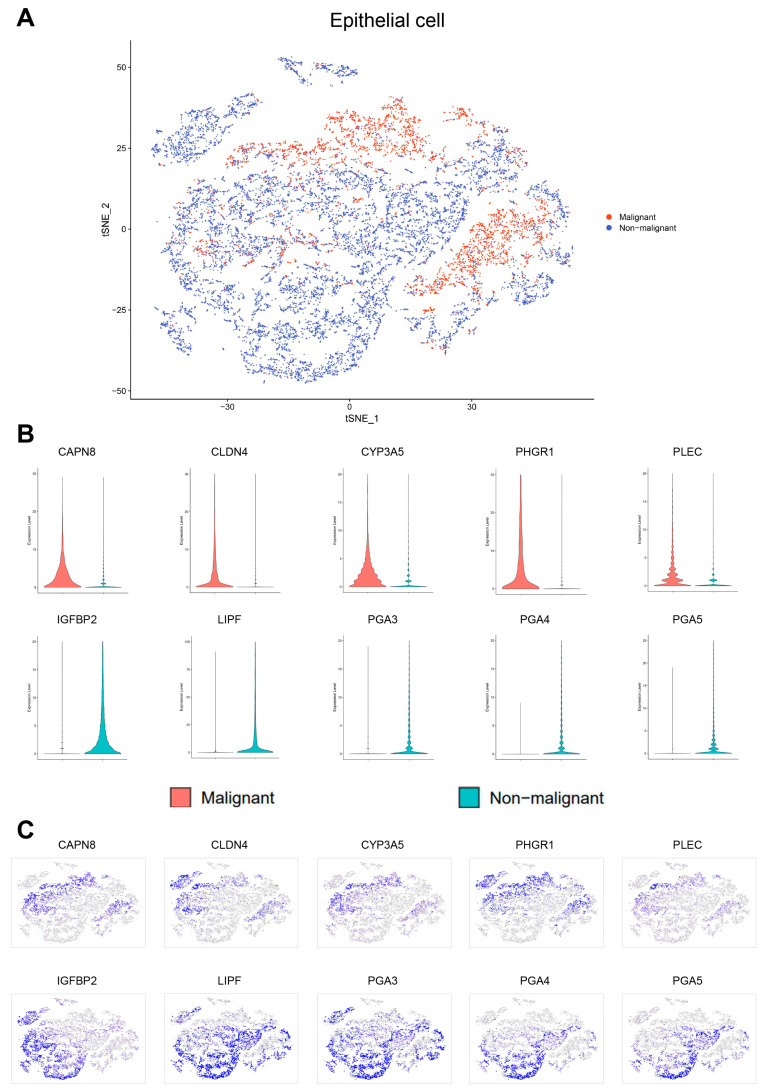
Classification of the epithelial cells as malignant or non-malignant. (**A**) tSNE plot showing the classification of malignant and non-malignant cells. (**B**) Violin plots showing the expression of ten representative genes with differential expression between malignant and non-malignant cells. (**C**) tSNE plots showing the expression of ten representative genes with differential expression.

**Figure 5 cells-11-02550-f005:**
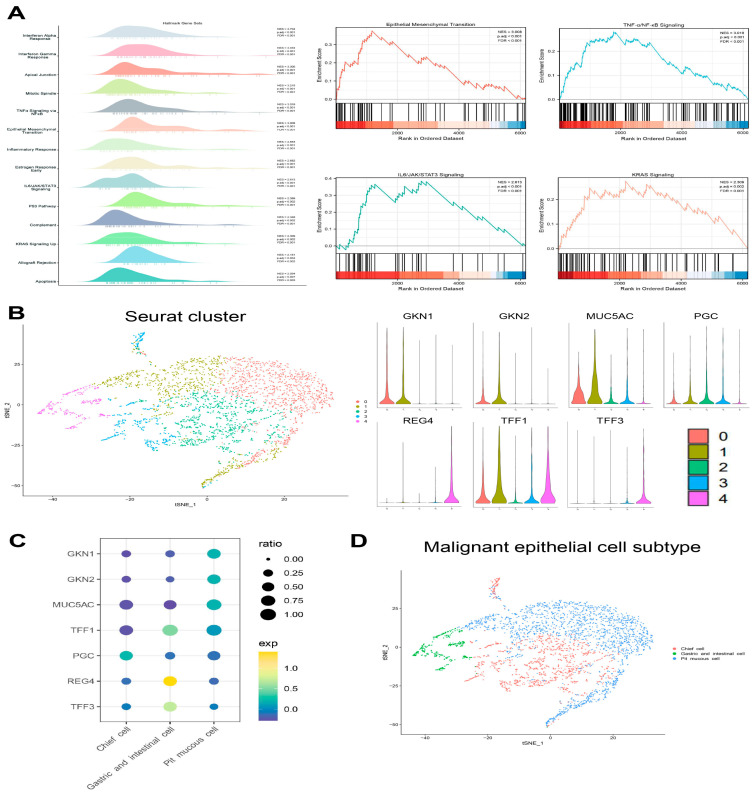
Cellular atlas of the malignant epithelial cell subtypes. (**A**) The GSEA results show the enrichment gene sets in malignant epithelial cells, and four of them were GC-associated gene sets. (**B**) tSNE plot showing the malignant epithelial cells seurat cluster, and Violin plots showing the expression distribution of subtype marker genes in five clusters. (**C**) Bubble plot showing the expression levels of marker genes in three subtypes. (**D**) tSNE plot showing the subtypes for malignant epithelial cells.

**Figure 6 cells-11-02550-f006:**
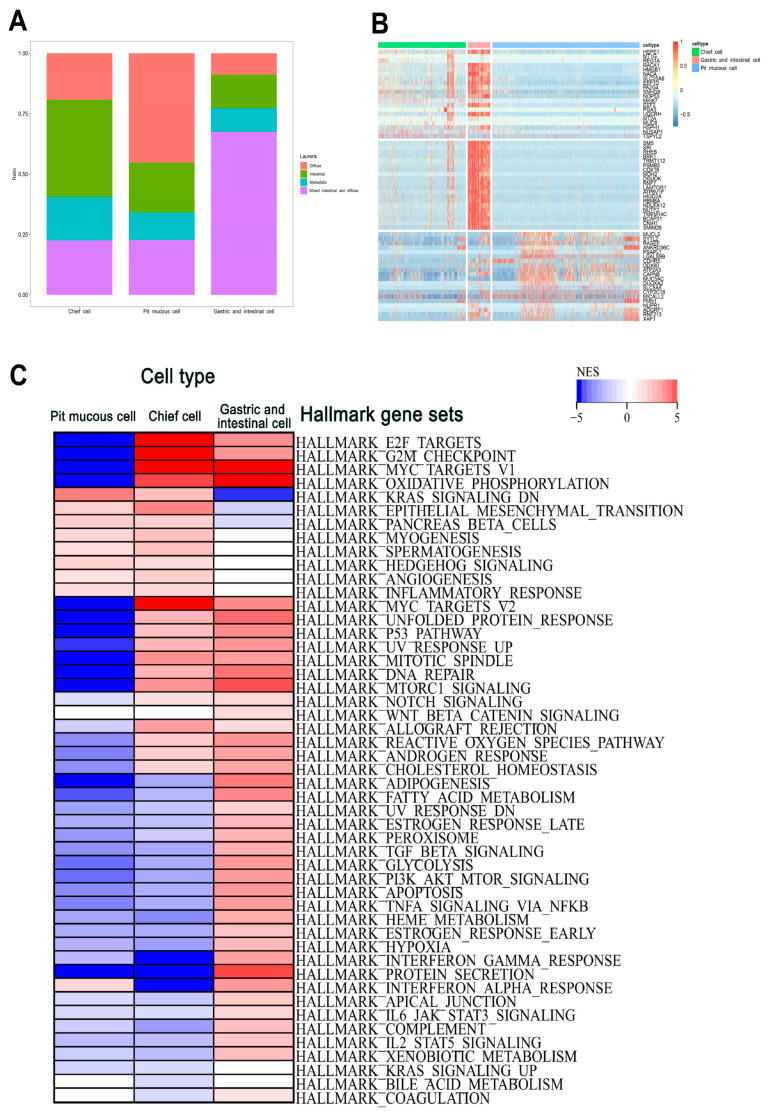
The heterogeneity of the malignant epithelial cell subtypes in GC. (**A**) The proportion of each subtype in Lauren’s histopathology. (**B**) Heatmap showing the top 20 marker genes in three subtypes. (**C**) The GSEA results showing the enrichment gene sets in three subtypes.

**Figure 7 cells-11-02550-f007:**
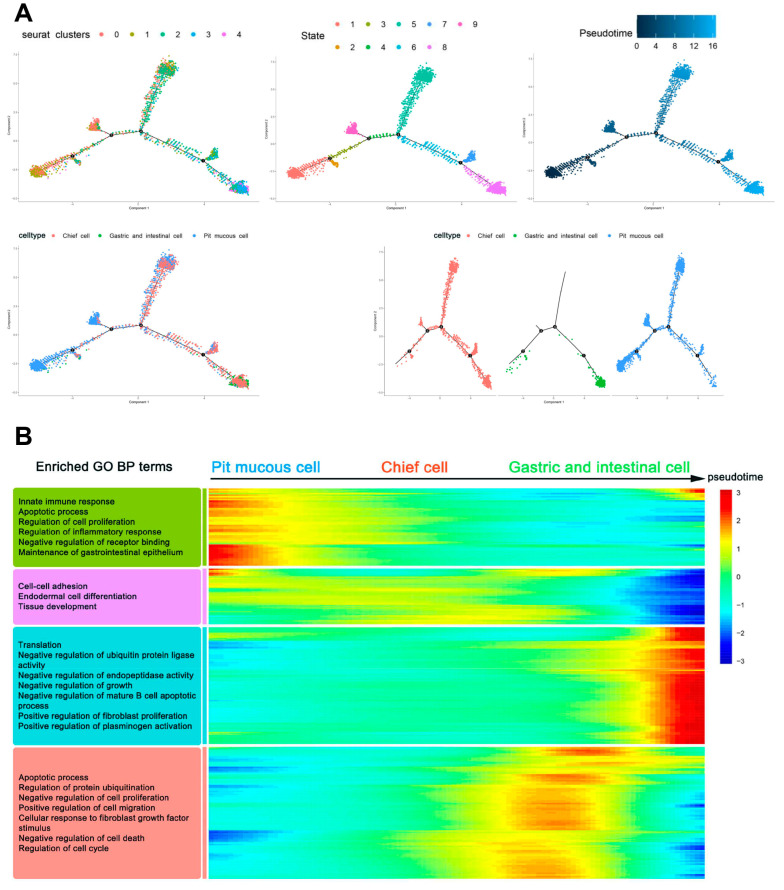
Trajectory analysis of malignant epithelial cells. (**A**) Trajectory plots showing the differentiation of the three subtypes. (**B**) Heatmap showing the scaled expression of dynamic genes along the pseudotime. Rows of the heatmap represent genes that show dynamic changes along the pseudotime, and these genes were clustered into four groups according to their expression pattern along the pseudotime. The annotated GO:BP terms in each cluster were provided.

**Figure 8 cells-11-02550-f008:**
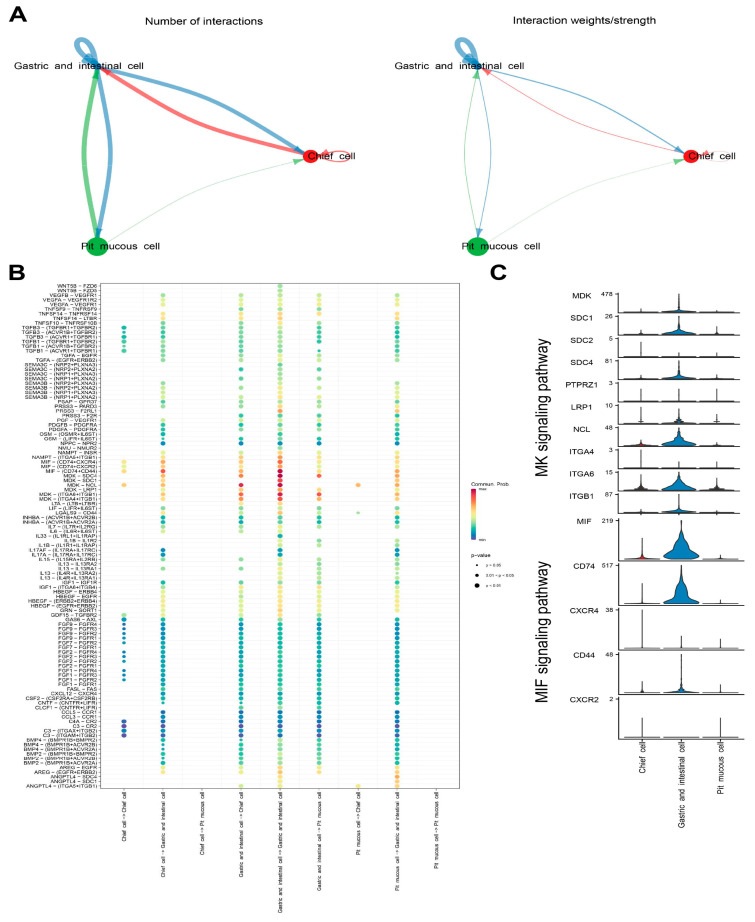
Cell–cell communication analysis of malignant epithelial cells. (**A**) Capacity for intercellular communication among the three malignant epithelial cell subtypes. The size of the circle is proportional to the number of cells. The cells that emit arrows express ligands and the cells that arrows point to express receptors. The more ligand–receptor pairs, the thicker the line. (**B**) Bubble plot showing the intercellular interaction relationship mediated by ligand–receptor pairs. (**C**) Violin plots showing the expression of representative genes in the MK and MIF signaling pathway.

**Figure 9 cells-11-02550-f009:**
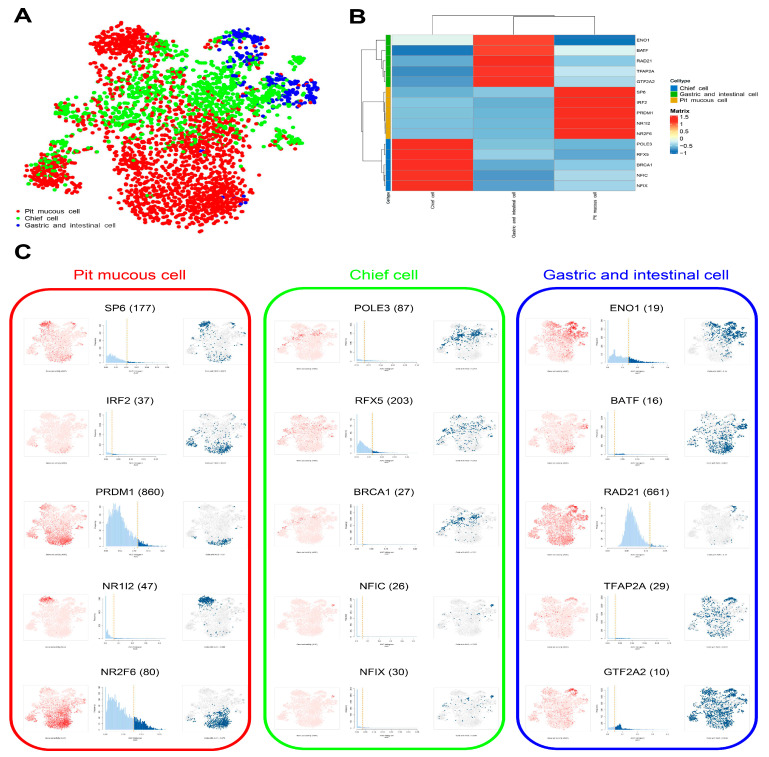
The SCENIC analysis predicted the TF. (**A**) tSNE plot showing the TF in three subtypes of malignant epithelial cells. (**B**) Heatmap showing the top 5 TF in three subtypes of malignant epithelial cells. (**C**) tSNE plots and histograms show the top 5 TF activities in three subtypes of malignant epithelial cells.

**Figure 10 cells-11-02550-f010:**
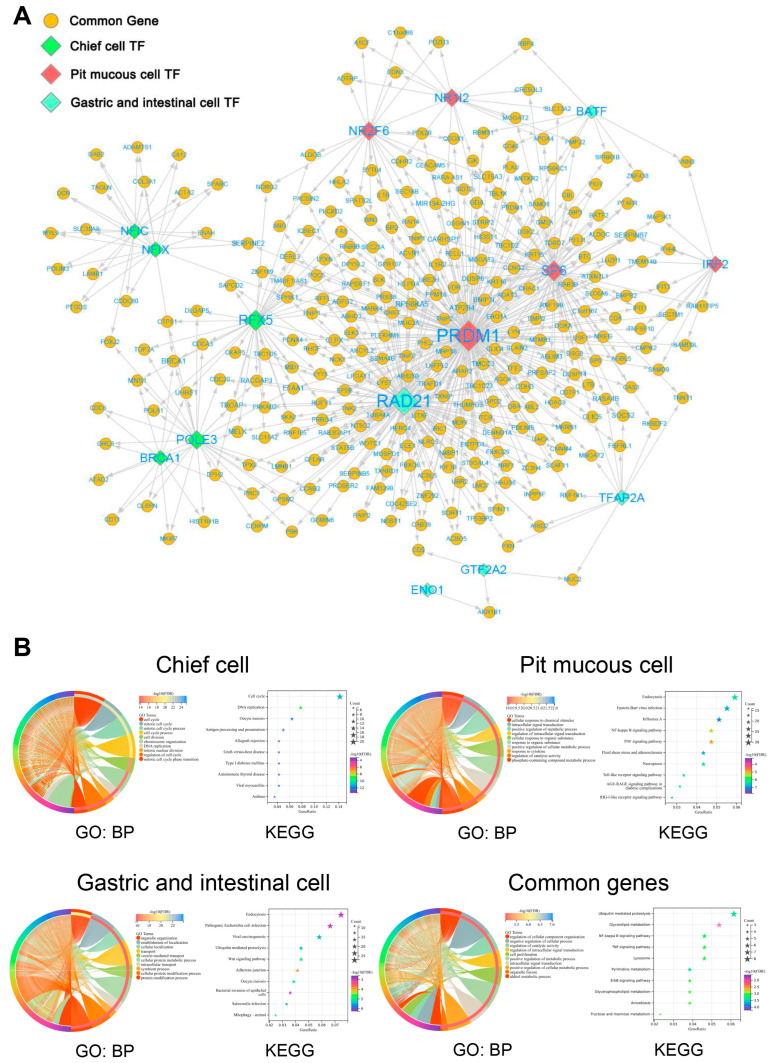
The TF-gene regulatory network of malignant epithelial cells. (**A**) Network showing the top 5 TF and their regulated genes in three subtypes of malignant epithelial cells. (**B**) Chord and bubble plots show the functional enrichment results of the top 5 TF regulated genes in three subtypes of malignant epithelial cells.

**Figure 11 cells-11-02550-f011:**
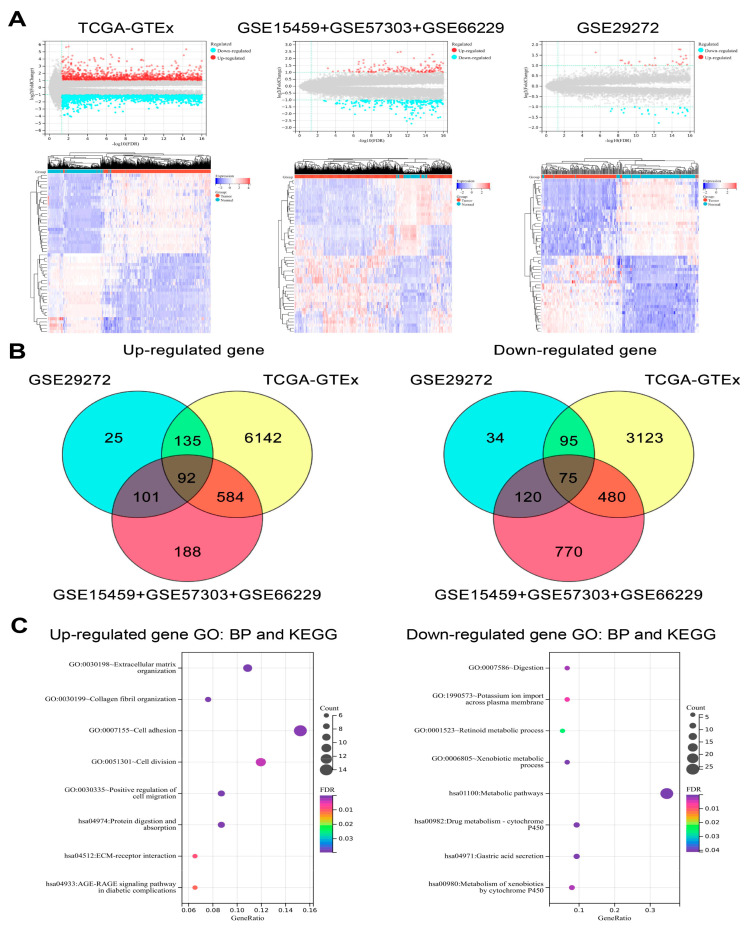
The analysis of the bulk RNA-seq datasets. (**A**) Volcano plots and heatmaps show the DEGs between the tumor and normal tissues in bulk RNA-seq datasets. (**B**) Venn plots showing the common up-regulated and down-regulated genes in different bulk RNA-seq datasets. (**C**) Bubble plots show the functional enrichment results of the up-regulated and down-regulated genes.

**Figure 12 cells-11-02550-f012:**
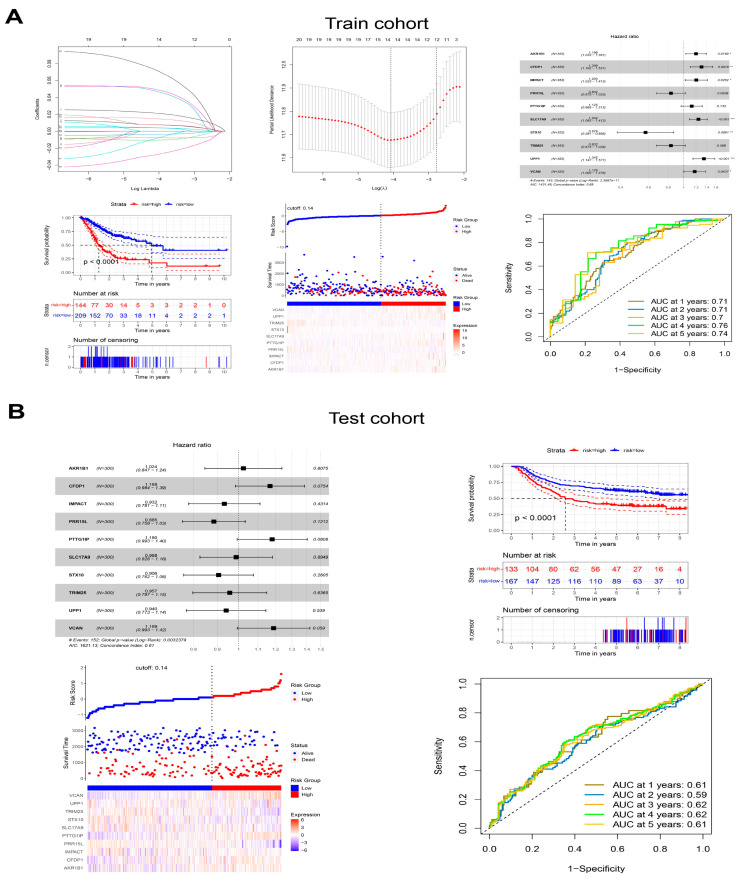
Generation and validation of the prognostic signatures in malignant epithelial cells. (**A**) The generation of the prognostic signatures in the training cohort. (**B**) The validation of the prognostic signatures in the test cohort. * *p* < 0.05, ** *p* < 0.01, *** *p* < 0.001.

## Data Availability

The datasets presented in this study can be found in online repositories. The names of the repository/repositories and accession number(s) can be found in the article/Appendix A.

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
