# Peer review of "Single-Cell and Bulk RNA Sequencing Reveal Malignant Epithelial Cell Heterogeneity and Prognosis Signatures in Gastric Carcinoma"

_cells, 2022, doi:10.3390/cells11162550_

Round 1

Reviewer 1 Report

The paper "Single-cell and bulk RNA sequencing reveals malignant epithelial cells heterogeneity and prognosis signatures in gastric carcinoma" by Huang and colleagues is an interesting reanalysis of published single-cell and bulk RNA-Seq gastic cancer data data. The authors identify different histologies of tumor progression, and provide ample support to their claim, both in terms of depth of results and quantity of independent data analyzed. Despite the evident value of their work, I have one major point to raise, and a few minor points:

- The authors should provide a complete signature of gastric cancer vs. normal tissue, one that is fully usable by other researchers. Supplementary Table 5.1, while providing markers for clusters in terms of most variable genes, is not what I mean. The most useful form of a signature is a transcriptome-wide table providing the differential statistics, Log2Fold Change, and p-value of every gene, in every comparison performed by authors on Gastric Cancer vs. normal tissue, in single-cell, bulk data, and each of the three subtypes revealed by the authors. This would support the true breakthrough of this paper (the discovery of heterogenous tumor subtypes in Gastric Cancer) and allow other researchers to detect the strength of such signatures in future datasets (also, possibly, beyond gastric cancers, and into other adenocarcinomas).

Minor points

- Line 69: The package Seurat is mispelled

- Line 72: the TCGA stomach adenocarcinoma dataset should be properly cited here as well (citation 5, or https://www.nature.com/articles/nature13480)

- Line 114: the R Monocle package should be cited when its use is clearly mentioned, specifically at the end of the sentence, not before (citation 16)

- Line 145: the R software as a whole should be properly cited (there is a recent review on R for bioinformatics and data science in the MDPI Life journal https://pubmed.ncbi.nlm.nih.gov/35629316/)

- Figure 2C - the total number of cells should be reported (e.g. on the right axis) for each of samples in this stacked bar plot (since the x-axis reports proportion of cells and not absolute number of cells)

Author Response

Thank you for your suggestions and comments. We have revised them according to the requirements and provided a transcriptome-wide table for other researchers. In addition, we invited a English-speaking expert to further refine and polish the language of the manuscript.

Q1: Line 69: The package Seurat is misspelled

A1: Thank you for your suggestion, we have revised it.

Q2: Line 72: the TCGA stomach adenocarcinoma dataset should be properly cited here as well (citation 5, or https://www.nature.com/articles/nature13480)

A2: Thank you for your suggestion, we have revised it.

Q3: Line 114: the R Monocle package should be cited when its use is clearly mentioned, specifically at the end of the sentence, not before (citation 16)

A3: Thank you for your suggestion, we have revised it.

Q4: Line 145: the R software as a whole should be properly cited (there is a recent review on R for bioinformatics and data science in the MDPI Life journal https://pubmed.ncbi.nlm.nih.gov/35629316/)

A4: Thank you for your suggestion, we have added citation.

Q5: Figure 2C - the total number of cells should be reported (e.g. on the right axis) for each of samples in this stacked bar plot (since the x-axis reports proportion of cells and not absolute number of cells)

A5: Thanks for your suggestion, we have added the number of cells in different samples to the figure.

Reviewer 2 Report

The authors present data from a total of 49994 cells sourced from nine patients with paired primary tumor and normal tissues to establishe a comprehensive single-cell transcriptional atlas to identify the cellular heterogeneity in malignant epithelial cells of GC using single-cell RNA sequencing (scRNA- 20 seq). Overall, the study aimed at providing a resource for understanding the 30 heterogeneity of GC that will contribute to accurate diagnosis and prognosis. With gastric carcinoma being a relevant neoplasia bearing significance all over the world this paper adds important and valuable information to the scientific community.

Minor comment to be addressed before publication encompass:

1. The title should be modified: Single-cell and bulk RNA sequencing reveal malignant epithelial cell heterogeneity and prognosis signatures in gastric carcinoma

2. In the introduction section, ll 46-47 should be changed to “High-throughput sequencing technologies, such as bulk RNA sequencing (bulk RNA-seq), have greatly accelerated…

3. In the results section: l 153 “after removal of low quality cells”, low quality cells should be defined in one sentence

4. In the results section: l 179 enriched for signaling pathways, any information on pathways regarding immunoevasion, such as PD-1L and Galectin-9?

5. In the results section l229: Do the authors detect gene signatures regarding immunoevasion (PD-1L and Galectin-9)?

6. Discussion section: Please add a paragraph describing potential limitations of this study (e.g. low patient number..)

7. Discussion section: The authors mention that all results of this study need to validated in experimental and clinical practice. Please outline this validation in a short paragraph of the Discussion section

8. Discussion section: A paragraph emphasizing more the potential clinical application and clinical benefit of results presented here would boost the significance of this paper. 

Author Response

Thank you for your suggestions and comments. We have revised them according to the requirements. In addition, we invited a English-speaking expert to further refine and polish the language of the manuscript.

Q1: The title should be modified: Single-cell and bulk RNA sequencing reveal malignant epithelial cell heterogeneity and prognosis signatures in gastric carcinoma

A1: Thank you for your suggestion, we have revised it.

Q2: In the introduction section, ll 46-47 should be changed to “High-throughput sequencing technologies, such as bulk RNA sequencing (bulk RNA-seq), have greatly accelerated…

A2: Thank you for your suggestion, we have revised it.

Q3: In the results section: l 153 “after removal of low quality cells”, low quality cells should be defined in one sentence

A3: Thank you for your suggestion, we have defined “low quality cells” in the article.

Q4: In the results section: l 179 enriched for signaling pathways, any information on pathways regarding immunoevasion, such as PD-1L and Galectin-9?

A4: The enrichment results were not enriched in the immunoevasion pathways.

Q5: In the results section l229: Do the authors detect gene signatures regarding immunoevasion (PD-1L and Galectin-9)?

A5: We did not detect any gene signatures associated with immuneevasion.

Q6: Discussion section: Please add a paragraph describing potential limitations of this study (e.g. low patient number..)

A6: Thank you for your suggestion, we have add a paragraph describing potential limitations of this study.

Q7: Discussion section: The authors mention that all results of this study need to validated in experimental and clinical practice. Please outline this validation in a short paragraph of the Discussion section

A7: Thank you for your suggestion, we have outlined this validation in a short paragraph of the Discussion section.

Q8: Discussion section: A paragraph emphasizing more the potential clinical application and clinical benefit of results presented here would boost the significance of this paper. 

A8: Thank you for your suggestion, we have emphasized more the potential clinical application and clinical benefit of results to boost the significance of this paper.